# Mindsponge-Based Reasoning of Households' Financial Resilience during the COVID-19 Crisis

**Minh-Hoang Nguyen** [1] , **Quy Van Khuc** [2] , **Viet-Phuong La** [1] , **Tam-Tri Le** [1] , **Quang-Loc Nguyen** [3] ,
**Ruining Jin** [4] , **Phuong-Tri Nguyen** [5] **and Quan-Hoang Vuong** [1,*]

1   Centre for Interdisciplinary Social Research, Phenikaa University, Yen Nghia Ward, Ha Dong District, Hanoi 100803, Vietnam
2   Faculty of Development Economics, VNU University of Economics and Business, Vietnam National University, Hanoi 100000, Vietnam
3   SP Jain School of Global Management, Lidcombe, NSW 2141, Australia
4   Civil, Commercial and Economic Law School, China University of Political Science and Law, Beijing 100088, China
5   Securities Research and Training Center, State Security Commission, Ho Chi Minh 700000, Vietnam
*   Correspondence: hoang.vuongquan@phenikaa-uni.edu.vn

**Abstract:** The COVID-19 crisis was remarkable because no global recession model could predict or provide early notice of when the coronavirus pandemic would happen and damage the global economy. Resilience to financial shocks is crucial for households as future crises like COVID-19 are inevitable. Therefore, the current study aims to examine the effects of financial literacy and accessibility to financial information on the financial resilience of Vietnamese households through the lens of an information-processing perspective. The Bayesian Mindsponge Framework (BMF) analytics was employed on a dataset of 839 samples for the investigation. We found that households of respondents with better financial knowledge and investment skills are less likely to be financially affected during the peak of the COVID-19 crisis, but the effect of investment skills is weakly reliable. Accessibility to financial information through informal sources (having a household member working in the financial sector) and formal sources (participating in a financial course) is positively associated with the respondents' financial knowledge and investment skills. This finding suggests that the spillover effect of financial knowledge and skills among residents exists, leading to better resilience toward financial shocks. However, if the financial information is inaccurate, it might lead to misinformation, false beliefs, and poor economic decisions on a large scale.

**Keywords:** financial literacy; financial resilience; information accessibility; mindsponge theory; crisis

## 1. Introduction

The COVID-19 pandemic has severely affected countries' socioeconomic development worldwide, especially one of the most important actors in any economy: households. Households sell their land, labor, capital, and entrepreneurial activities in exchange for income and use the proceeds to buy the goods and services they need. These activities create the flow of money within the economy. However, the flow of money was disrupted as a large number of households became unemployed, and household-owned businesses were damaged due to lockdowns, social distancing, and non-essential business closures (Akkermans et al. 2020; Kuckertz et al. 2020). Evidently, the pandemic shock is found to cause unemployment, significant losses in household-owned business income, and vulnerability perception, which eventually leads to a higher likelihood of liquidity constraints, increased saving and investing willingness, and changing consumer behaviors (Li et al. 2020; Achou et al. 2020; Yue et al. 2021; Yazdanparast and Alhenawi 2022). As entrepreneurs experienced downward development trends, it subsequently continued to result in rising levels of unemployment.

Financial resilience is the ability to handle adverse economic outcomes, especially unexpected ones, and to bounce back swiftly from tough financial times (McKnight and Rucci 2020; Sakyi-Nyarko et al. 2022). In addition to overcoming financial adversity, financial resilience is also associated with various other critical outcomes, including household debt, emotional well-being, educational performance, family stability, and reliance on the government (McKnight and Rucci 2020). Thus, during the difficult financial conditions induced by the COVID-19 pandemic, households' financial resilience is crucial as it can lead to more effective resource allocation and higher financial stability at both the micro and macro levels.

Financial literacy has been advocated as one of the important determinants of financial resilience, as it helps individuals generate rational and well-informed decisions about saving, spending, and investment (Lusardi et al. 2021; Klapper and Lusardi 2020; Lusardi and Mitchell 2014). Furthermore, financial literacy facilitates financial inclusion—the availability and equality of opportunities to access various financial services—among households, especially disadvantaged groups in developing countries (Sahay et al. 2015; Kazemikhasragh et al. 2022; Cicchiello 2021). A better inclusion in financial services can support households to make productive investments in the face of risk and prepare for shocks through insurance and low-cost savings, consolidating their economic resilience (Kass-Hanna et al. 2022; Belayeth Hussain et al. 2019; Sakyi-Nyarko et al. 2022).

Several studies have provided empirical evidence for the relationship between financial literacy and financial resilience (Kass-Hanna et al. 2022; Erdem and Rojahn 2022). However, their reasoning approach mainly focuses on individuals' observable earning, saving, and investing patterns or behaviors. At the same time, little has been known about the underlying mechanism of such patterns or behaviors. To understand the underlying mechanism of financial literacy and resilience, mindsponge-based reasoning is a potential approach.

The mindsponge mechanism is an information-processing theory that explains how an individual's mind can influence their mental processes and behaviors and how the mind can be reinforced, modified, and shifted through the inflow and outflow processes of information (Vuong 2022a; Vuong and Napier 2015). The theory has been used as a theoretical framework for behavioral and psychological studies on multiple issues, such as innovations (Nguyen 2022; Vuong 2022b), migration intention (Khuc et al. 2022; Vuong et al. 2022c), suicidal ideation (Nguyen et al. 2021), risk aversion (Nguyen et al. 2022a), environmental psychology (Nguyen and Jones 2022a, 2022b), etc.

From a metaphysical perspective, everything within and beyond human minds can be examined in terms of information (Davies and Gregersen 2014; Adriaans 2020; Graziano 2022). Therefore, an information-based approach will provide greater versatility in explaining and reasoning how financial literacy can improve financial resilience and how it can be improved through access to financial information. This type of reasoning is essential in the digital age. In the wake of digitalization, improving households' financial literacy and inclusion through digital platforms is one prominent pathway to enhance financial resilience (Kass-Hanna et al. 2022; Cicchiello 2020; Gabor and Brooks 2017). The digital infosphere is a multiplex and dynamic environment with many communication and information dissemination modes. Within such an environment, the pattern- or behavior-based reasoning of financial literacy and resilience (Salignac et al. 2019; Lusardi and Mitchell 2014) appears less effective in explaining emerging phenomena because individuals are exposed to more information sources, types, and reliability levels. Especially the digital platform was the major channel for communication and information dissemination during the COVID-19 pandemic as a result of lockdown and social distancing (Minh et al. 2021).

Understanding how households could be financially resilient during the COVID-19 pandemic can offer lessons for improving a country's socioeconomic sustainability when facing an upcoming crisis. A future pandemic is inevitable due to the increasing number of zoonoses (infectious diseases that jump from animals to humans), making lessons from the COVID-19 pandemic even more valuable (Smith 2021). As a developing country, Vietnam

had successfully fended off the COVID-19 impacts in the first few waves. However, starting on 27 April 2021, the fourth wave of COVID-19, in addition to causing heavy casualties (Minh et al. 2021), severely damaged Vietnam's economy. As a result, the Gross Domestic Product (GDP) growth rate in the nine-month of 2021 dived from 7% to 1.42%, and domestic unemployment and underemployment rates surged (Le and Lam 2021). Therefore, insights regarding the effectiveness of households' financial literacy on financial resilience during the fourth pandemic wave will help policymakers acquire better preparations for upcoming crises (Vuong 2018). Moreover, the relationship between households' financial literacy and financial resilience remains underresearched in Vietnam.

For these reasons, the current study aims to examine how financial literacy and accessibility to financial information can influence the financial resilience of Vietnamese households during the COVID-19 pandemic through the lens of an information-processing perspective. The Bayesian Mindsponge Framework (BMF) analytics, with the mindsponge mechanism used to construct theoretical models and Bayesian inference employed to fit those models, was performed for investigation.

The paper comprises five main sections. The Section 1 explains the rationale and goal of the current study. The Section 2 explains how financial resilience can be enhanced by improving financial literacy and accessibility to financial information. The Section 3 describes the materials and methods employed for the statistical analysis, while the Section 4 presents the analysis's results. The Section 5 discussed the findings with existing literature, implications, future research directions, and limitations.

## 2. Mindsponge-Based Reasoning

Resilience is a concept that originated from ecology and was later developed and applied in other fields of study, such as behavioral and psychological sciences (Norris 2011; Buikstra et al. 2010), mental health research (Southwick et al. 2014), economics (Pant et al. 2014), and disaster management (Tadele and Manyena 2009). Recently, the resilience concept has been introduced to financial studies and is often referred to as "financial resilience" (Salignac et al. 2016, 2019). Based on prior studies of resilience in other fields, Salignac et al. (2019) summarized four main features of an individual's resilience and used them to define financial resilience. Those four main features are: (1) obtaining knowledge about the adversity, (2) obtaining the ability to predict the risk associated with such adversity, (3) obtaining accessibility and knowledge of available alternatives, and (4) obtaining the resources to adapt successfully. As such, financial resilience can be defined as "an individual's ability to access and draw on internal capabilities and appropriate, acceptable and accessible external resources and supports in times of financial adversity" (Salignac et al. 2019). In other words, it is an individual's ability to manage internal and external financial resources to cope with and recover from the financial crisis.

Here, we employed the mindsponge mechanism to redefine financial resilience and financial literacy and explain how financial resilience can be formed through the information-processing approach. From the mindsponge approach, knowledge and ability are outcomes of mental processes occurring within the mind (Nguyen et al. 2022c; Vuong et al. 2022a). While knowledge is deemed information stored within the mind, the ability can be considered the mind's capability of capitalizing on information existing within the mind for subsequent thinking processes or behaviors. Therefore, knowledge and ability are bounded by the amount and types of information stored within the mind and permeated from the environment. Following this way of thinking, financial resilience can be defined as the individual's capability to manage information within the mind and information absorbed from the external environment to deal with and recover from financial adversities. As a result, the capability depends on the amount and types of information related to financial issues that are stored within the mind and the mind's capability to process information related to financial issues. If the information is related to financial issues, it can be considered financial literacy. This reasoning does not contradict Lusardi and Mitchell (2014) behavior-based definition, which refers to financial literacy as people's "ability to process economic

information and make informed decisions about financial planning, wealth accumulation, debt, and pensions". Rather, information-based reasoning provides more flexibility in explaining the flow of financial knowledge.

Specifically, financial information within the mind is used as inputs for mental processes to generate expectations of financial risks (or costs) and alternatives (or benefits), facilitating decision-making. Moreover, through the updating mechanism, better financial information within the mind will also improve the subsequent information-seeking, evaluating, and filtering processes to maximize the perceived benefits and minimize the perceived costs of the individuals. For example, people with good financial abilities perform better in wealth accumulation, retirement planning, and saving (Lusardi and Mitchell 2011; Behrman et al. 2010). They are also more likely to participate in financial markets and stock investments (both directly and indirectly through mutual funds and retirement accounts) (Almenberg and Dreber 2015; Christelis et al. 2010). Furthermore, studying 26,000 respondents in the United States, Lursadi and Scheresberg find that most high-cost borrowers have very poor levels of financial literacy, which means they lack numeracy and a grasp of basic financial concepts (Lusardi and Scheresberg 2013).

Therefore, we postulated that people with greater financial literacy (financial knowledge and investment skills) are more likely to have wiser decision-making regarding financial issues, which not only helps them have better preparation before the crisis but also leads to better financial outcomes during the crisis. To test this assumption, the following two research questions (RQs) were proposed:

RQ1: Were households with better general financial knowledge less likely to be financially affected during the peak of the COVID-19 pandemic?

RQ2: Were households with better investment skills less likely to be financially affected during the peak of the COVID-19 pandemic?

Then, how can financial knowledge and investment skills be improved? One of the fundamental principles of the mindsponge mechanism is that "an information particle needs to exist in the environment (objective world) and locate within the perceivable range to be absorbed into the mind (subjective world)". In other words, Vietnamese households need access to financial information to develop their financial knowledge and investment skills. The information sources can be from any people and organizations within the perceivable ranges (or the physical range within which a person can see, hear, or become aware of something through the senses) of the households, regardless of its formality.

Thus, we continued to propose the following four research questions to examine the relationship between information accessibility and financial knowledge/skills:

RQ3: Were households accessing informal information sources more likely to acquire better financial knowledge?

RQ4: Were households accessing informal information sources more likely to acquire better investment skills?

RQ5: Were households accessing formal information sources more likely to acquire better financial knowledge?

RQ6: Were households accessing formal information sources more likely to acquire better investment skills?

Here, the formal source refers to the participation in a training program or course about financial issues, while the informal source refers to knowing someone that has expertise about financial issues.

## 3. Materials and Methods

### 3.1. Materials

The study was designed to use primary data. We have undergone many steps following Bailey (Bailey 1994) to collect this data. In the first step, we formed a focus group. The main task of the focus group is to help revise the questionnaire until it is completed based on the ask-feedback process. Because the direct interview method was not used in

this study, a well-designed questionnaire is essential for ensuring the data's validity and reliability. This is due to the Vietnam government's strict application of many measures such as social distancing, social isolation, and movement restrictions prevented the team from conducting a direct survey.

In the second step, a pilot survey was conducted to improve the quality of the questionnaire, evaluating the benefits and drawbacks of the survey to increase the rate of respondents and supplement the hypothesis if possible. This step was completed successfully, as expected, which aided in completing the questionnaire. The finalized questionnaire is divided into four sections: household financial security perceptions, household financial management skills and satisfactions, and personal information.

In the third step (final survey), the respondents were randomly selected and surveyed during June and July 2021. That was when Vietnam encountered the fourth wave, which recorded a spike in confirmed cases daily, with a peak of 6000 cases (https://covid19.gov.vn/ accessed on 13 November 2022). We selected this period for survey collection because economic activities were paralyzed, and many people suffered tremendously from such an economic backlash.

We used many communication methods such as Viber, Zalo, and phone to maintain mutual interaction to solve any issues arising during the data collection. We did not restrict the study locations for two reasons. First, we collect data using an online combined phone method that can easily help collectors reach a large number of respondents. Second, in addition to sample size and randomness consideration, data representativeness is far more important. In this regard, the larger the survey area, the more representative the data, resulting in a stronger conclusion associated with high generalization. The survey team completed the survey with 850 responses. After data refinement, 839 out of 850 valid responses remained, a sufficient sample size for further data processing and analysis (VanVoorhis and Morgan 2007).

*3.2. Analytical Approach*

This study employed the Bayesian Mindsponge Framework (BMF) analytics, which combines the strengths of the mindsponge mechanism and Bayesian inference for constructing and fitting models. One of the key advantages of BMF analytics is allowing researchers to construct models that follow the parsimony principle (entities should not be multiplied beyond necessity) (Nguyen et al. 2022b, 2022c). Parsimonious models are preferable because they aid the discovery of data patterns and the generation of more precise and integrated conclusions (Bentler and Mooijaart 1989; Cougle 2012; Simon 2001). Mindsponge mechanism, constructed based on set theory logic (assuming the mind is a set of information and a part of the larger set of information—environment), provides a conceptual framework to define the boundary of the studied subjects (Nguyen 2022). This helps achieve parsimony when constructing models. Meanwhile, Bayesian inference treats all the properties probabilistically, including unknown parameters and uncertainties (Gill 2014; McElreath 2018), so it is a suitable method for inferring parsimonious models.

The second major advantage of BMF analytics is prior incorporation. Prior incorporation allows researchers to integrate prior evidence, beliefs, or assumptions into the models to generate posterior distributions. If the prior distributions are specified appropriately, Bayesian analysis will result in a more precise estimation with a small sample size (Uusitalo 2007; McNeish 2016). In addition, incorporating priors into the analysis can reduce the risk of multicollinearity in statistical analysis (Leamer 1973; Jaya et al. 2019; Adepoju and Ojo 2018). However, this application of Bayesian inference is usually criticized for its potential bias due to the involvement of subjectivity. By applying the mindsponge theory for reasoning, the priors selected in the current study have received theoretical support using set theory logic, reducing potential biases (Nguyen et al. 2022b).

In the current study, three models were constructed to seek answers to the aforementioned research questions. Five variables were employed for the model construction: *FormalFinInfo, InformalFinInfo, GeneralFinKno, InvestmentSkill, and FinImpactCovid*

(see Table 1). Specifically, to answer RQs 1 and 2, Model 1 was constructed to examine whether financial knowledge and investment skills are positively associated with financial resilience during the COVID-19 pandemic.

$$FinImpactCovid \sim \alpha + GeneralFinKno + InvestmentSkill \tag{1}$$

Then, Models 2 and 3 were constructed and fitted to test whether accessibility to formal and informal financial sources are associated with the financial skills and financial knowledge of households, respectively:

$$InvestmentSkill \sim \alpha + FormalSource + InformalSource \tag{2}$$

$$GeneralFinKno \sim \alpha + FormalSource + InformalSource \tag{3}$$

The *FormalSource* variable was measured by asking, "Have you ever taken a financial course?" while the *InformalSource* variable was measured by asking, "Is there anyone in your house working in the financial sector (e.g., bank, commerce, accounting)?". The individual's financial knowledge and investment skills are proxied by self-reported satisfaction/confidence about their financial knowledge and financial skills, respectively.

**Table 1.** Variable description.

| Variable | Meaning | Type of Variable | Value |
|----------|---------|------------------|-------|
| *FormalSource* | Financial training | Binary | Yes = 1<br>No = 0 |
| *InformalSource* | Respondents' job in the field of bank/accounting/business | Binary | Yes = 1<br>No = 0 |
| *GeneralFinKno* | Respondents' satisfaction/confidence level towards their general financial knowledge (e.g., expenditure, borrowing, and investment policies) | Continuous | From 1 (very dissatisfied/unconfident) to 5 (very satisfied/confident) |
| *InvestmentSkill* | Respondents' satisfaction/confidence level towards their investment skills (e.g., saving, stock trading, real estate investment) | Continuous | From 1 (very dissatisfied/unconfident) to 5 (very satisfied/confident) |
| *FinImpactCovid* | COVID-19 pandemic's impact on the financial conditions of the households | Continuous | From 1 (very low) to 5 (very high) |

We used the bayesvl R package to analyze the data (Vuong et al. 2020, 2022b). The package is employed because of its easy-to-use operation and good visualization power. We set 5000 iterations (with 2000 iterations for the warm-up process) and four Markov chains for the simulation. The Pareto smoothed importance-sampling leave-one-out cross-validation (PSIS-LOO) method was employed to check the model's goodness of fit (Vehtari et al. 2017). The Markov property (or model convergence) was assessed using diagnostic statistics, like effective sample sizes (*n_eff*) and Gelman–Rubin shrink factors (*Rhat*) and trace plots.

As for prior selection, we fitted the models two times with different prior distributions. In the first simulation, we employed the prior distributions reflecting our beliefs in the associations between the predictor and outcome variables. For Model 1, the prior distributions were set as normal distributions with mean values at $-0.1$ and standard deviations at 0.05. For Models 2 and 3, prior distributions were set as normal distributions with mean values at 0.1 and standard deviations at 0.05. In the latter simulation, we employed the prior distributions reflecting our disbelief in the associations. For all three models, the prior distributions were set as a normal distribution with the mean value at 0 and the standard deviation at 0.05. If the patterns of the posterior distributions do not change significantly when the priors are changed, the estimations can be deemed robust.

## 4. Results

### 4.1. Model 1: Financial Resilience to COVID-19

The first model examines the effects of respondents' financial skills and literacy on the COVID-19 pandemic's impacts on their financial conditions. Because the *k* values of the PSIS diagnostic plot are less than 0.5, Model 1 can be deemed to fit well with the data (see Figure 1). This also implies that the parsimony principle is held as the model is not under-fitted (or oversimplified).

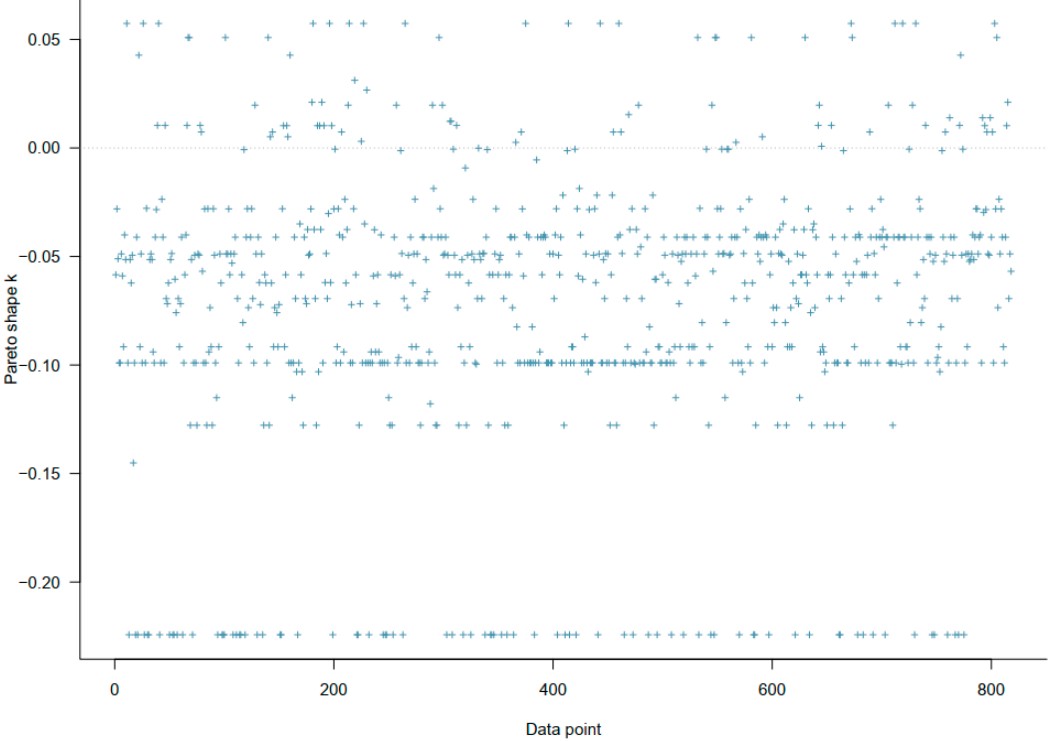

**Figure 1.** Model 1's PSIS diagnostic plot.

It can be seen from Table 2 that the *n_eff* values of all coefficients are greater than 1000, and *Rhat* values are equal to 1, signifying the model convergence. The trace plots also confirm Markov chains convergence. Specifically, Figure 2 displays the zig-zag fluctuation around a central equilibrium of coefficients' Markov chains. This is a good signal that the chains have converged to the same posteriors, so the simulated results can be used for later interpretation.

**Table 2.** Model 2's simulated posterior coefficients.

| Parameter | Priors Reflecting a Belief in the Association | | | | Priors Reflecting Disbelief in the Association | | | |
| --- | --- | --- | --- | --- | --- | --- | --- | --- |
| | **Mean** | **SD** | ***n_eff*** | ***Rhat*** | **Mean** | **SD** | ***n_eff*** | ***Rhat*** |
| *Constant* | 3.77 | 0.10 | 6652 | 1 | 3.65 | 0.09 | 6821 | 1 |
| *GeneralFinKno* | −0.07 | 0.03 | 6921 | 1 | −0.05 | 0.03 | 6517 | 1 |
| *InvestmentSkill* | −0.03 | 0.03 | 6342 | 1 | −0.01 | 0.03 | 7321 | 1 |

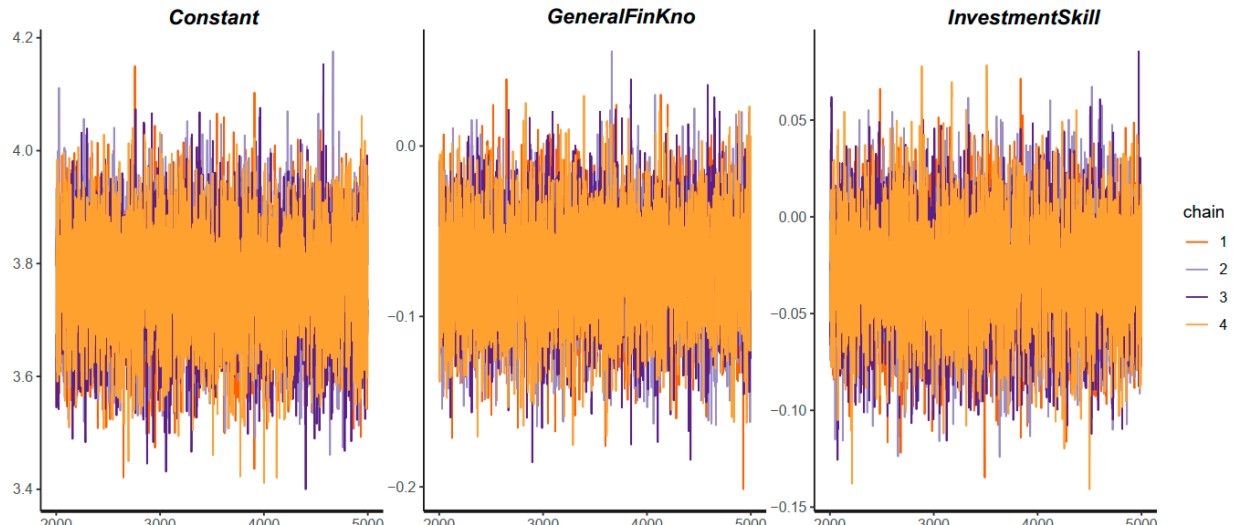

**Figure 2.** Model 1's trace plots.

It should be noted that we interpret the estimated results using priors reflecting our beliefs in the associations, while the estimated results using priors reflecting our disbelief in the associations are used to check the result's robustness. In Table 2, the simulated posterior results show that *GeneralFinKno* and *InvestmenSkill* have clear impacts on financial resilience. Both variables are found to be negatively associated with the *FinImpactCovid* variable ($\mu_{GeneralFinKno} = -0.07$ and $\sigma_{GeneralFinKno} = 0.03$; $\mu_{InvestmenSkill} = -0.03$ and $\sigma_{InvestmenSkill} = 0.03$). To elaborate, those with better financial skills and knowledge are less financially affected by the COVID-19 pandemic. All coefficients' distributions are demonstrated in Figure 3. Most of the posterior distributions of *GeneralFinKno* and *InvestmenSkill* coefficients are located on the negative side of the *x*-axis's origin. However, a portion of *InvestmenSkill*'s distribution is still located on the positive side of the *x*-axis's origin, indicating that *InvestmenSkill* has some probability of being positive.

After adjusting the priors to reflect our disbelief in the associations, the negative effects of *GeneralFinKno* and *InvestmenSkill* remain. Nevertheless, the negative effect of *InvestmenSkill* becomes less reliable as its absolute mean value is much lower than its standard deviation (0.01 compared to 0.03). Based on the posterior distributions shown in Figure 3 and the posterior results after adjusting prior distributions, it is conclusive that the effect of *GeneralFinKno* is strongly reliable, while that of *InvestmenSkill* is weakly reliable.

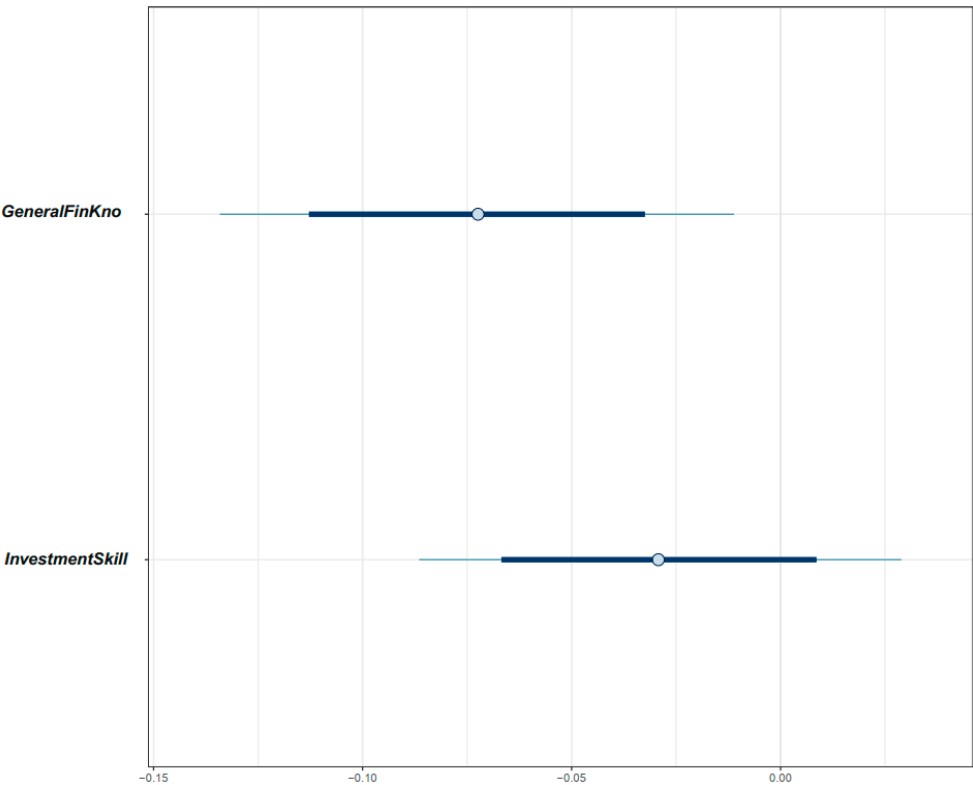

**Figure 3.** Model 1's posterior distributions on the interval plot.

### 4.2. Model 2: Financial Investment Skill

Model 2 examines whether respondents' accessibility to formal and informal financial sources can improve their financial investment skills. All $k$ values shown in Model 2's PSIS diagnostic plot are below 0.5, hinting at the model's high goodness of fit (see Figure 4).

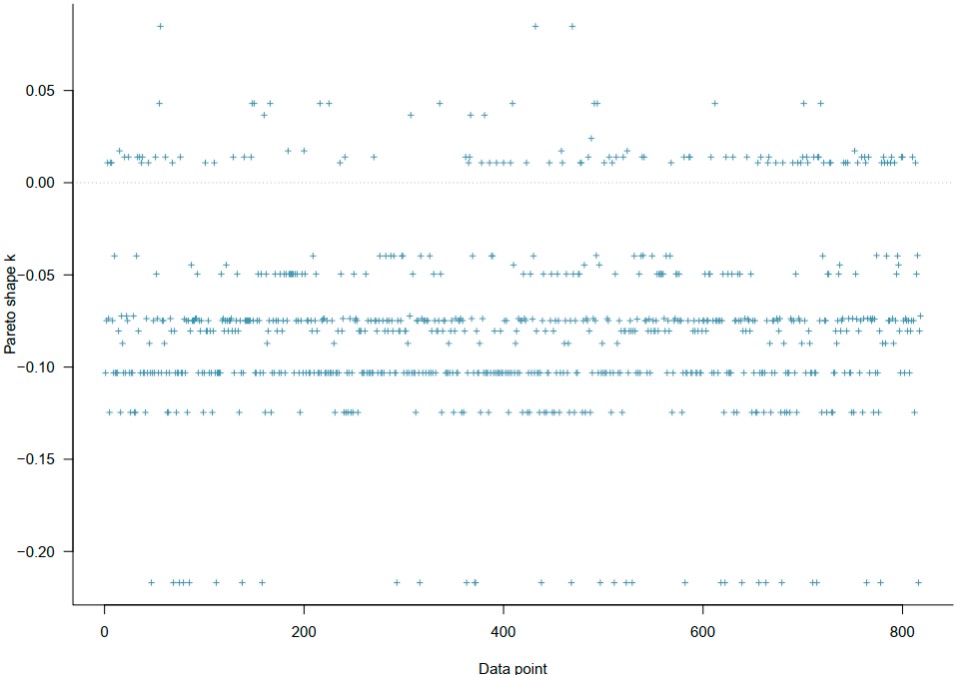

**Figure 4.** Model 2's PSIS diagnostic plot.

The trace plots for Model 2 are visualized in Figure 5, which shows that all Markov chains after the warm-up period (after the 2000th iteration) are "clean and healthy" and convergent at specific posterior values. The diagnostic statistics are also consistent with the convergence signals of trace plots, as all the *n_eff* values are greater than 1000, and *Rhat* values are equal to 1 (see Table 3).

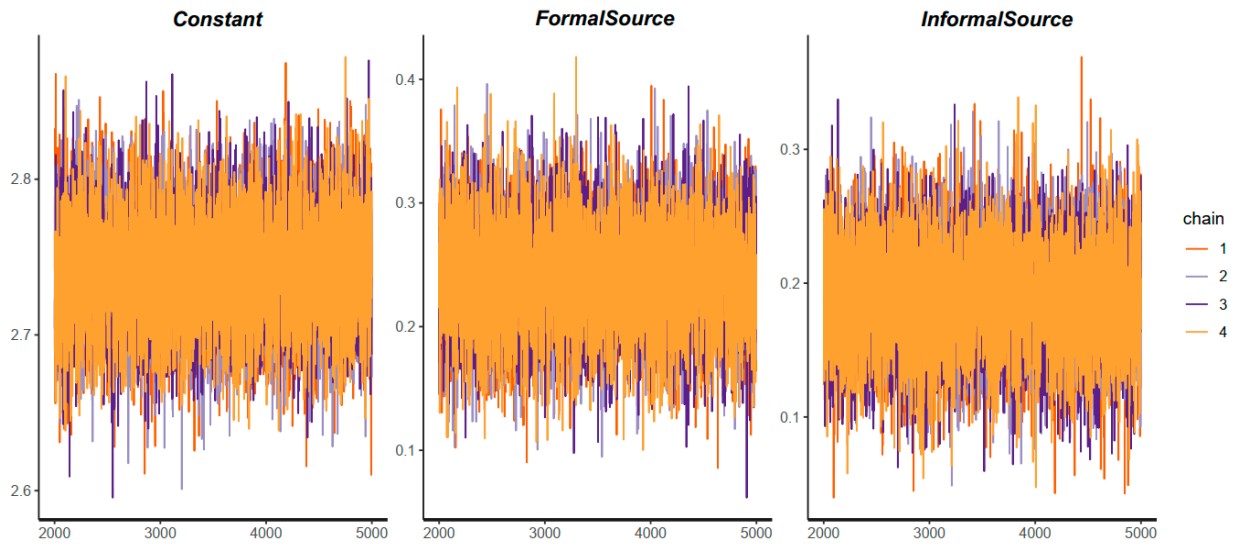

**Figure 5.** Model 2's trace plots.

**Table 3.** Model 2's simulated posteriors.

| Parameter | Priors Reflecting a Belief in the Association | | | | Priors refLecting Disbelief in the Association | | | |
|---|---|---|---|---|---|---|---|---|
| | **Mean** | **SD** | *n_eff* | *Rhat* | **Mean** | **SD** | *n_eff* | *Rhat* |
| *Constant* | 2.74 | 0.04 | 10,221 | 1 | 2.77 | 0.04 | 10,750 | 1 |
| *FormalFinInfo* | 0.24 | 0.04 | 11,202 | 1 | 0.17 | 0.04 | 10,425 | 1 |
| *InformalFinInfo* | 0.19 | 0.04 | 10,554 | 1 | 0.13 | 0.04 | 10,254 | 1 |

From the simulated posterior results of Model 2, being accessible to formal financial information sources (attending a financial course/training) is positively associated with a higher level of respondents' financial investment skills ($\mu_{FormalFinInfo}$ = 0.24 and $\sigma_{FormalFinInfo}$ = 0.04). The accessibility to the informal financial source (having a household member working in the financial sector) also leads to a positive contribution to the financial investment skills of respondents ($\mu_{InformalFinInfo}$ = 0.19 and $\sigma_{InformalFinInfo}$ = 0.04). Both posterior distributions of *FormalFinInfo* and *InformalFinInfo* fall entirely on the positive side of the *x*-axis (see Figure 6), and their positive effects remain robust even when prior distributions are modified (see Table 3). These signals imply the high reliability of the results. Notably, the positive effect of *FormalFinInfo* on *InvestmentSkill* seems to be greater than that of *InformalFinInfo* (see Figure 6).

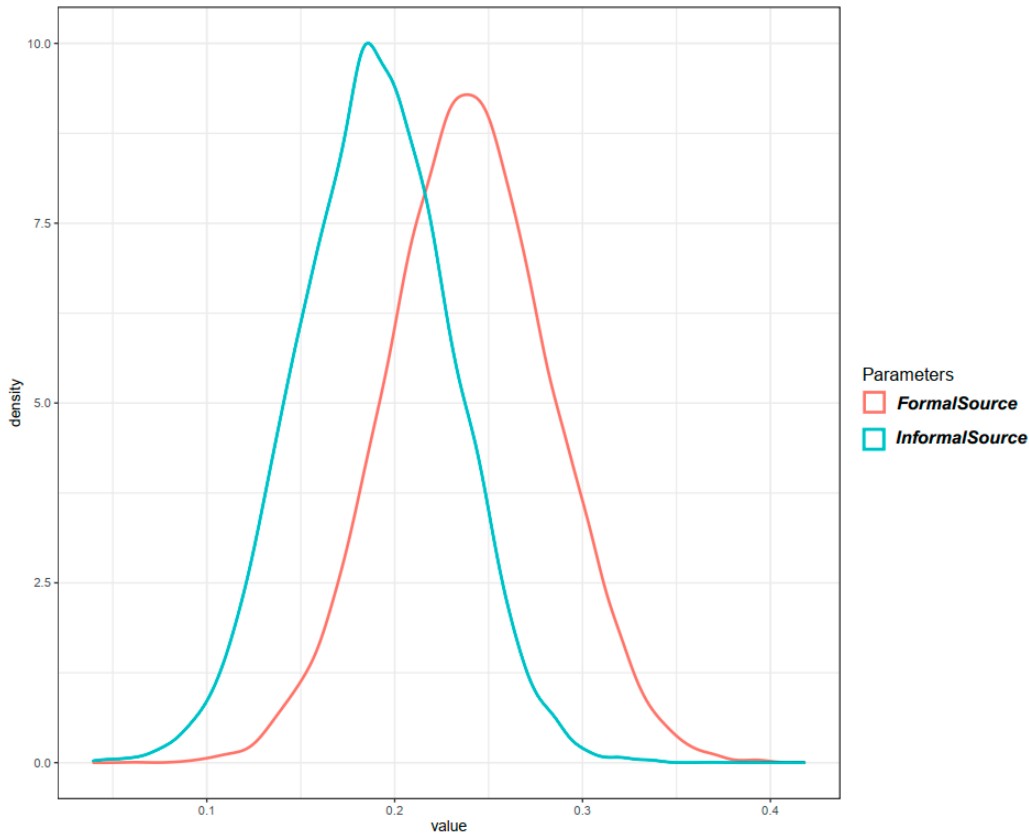

**Figure 6.** Model 2's posterior distributions on the density plot.

### 4.3. Model 3: General Financial Knowledge

The third model examines the associations between respondents' accessibility to formal and informal financial sources and general financial knowledge. The model's goodness of fit can be deemed acceptable because all *k* values on the PSIS diagnostic plot are below 0.5 (see Figure 7). In other words, the model is parsimonious but not oversimplified.

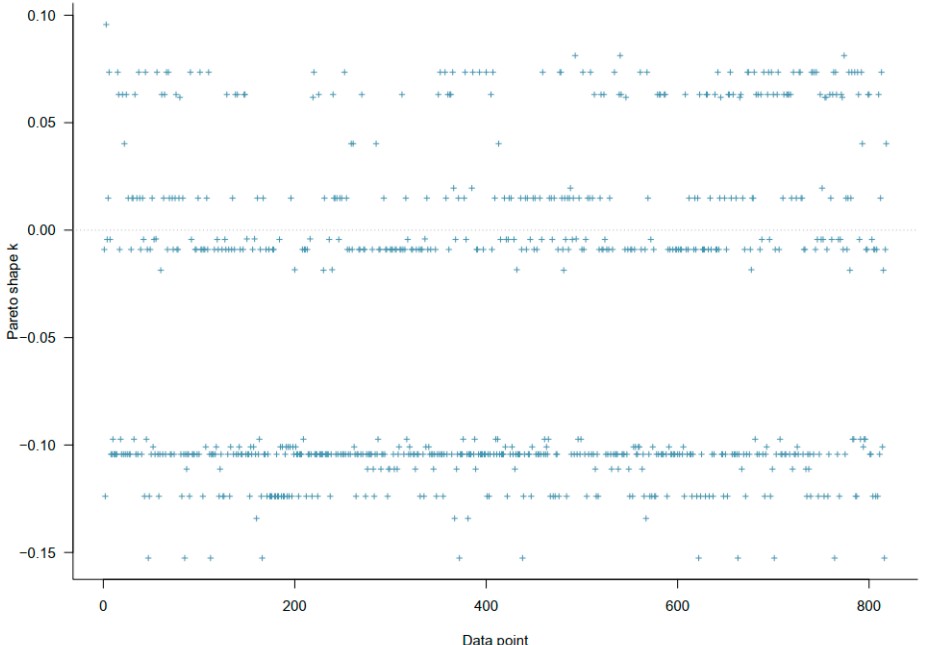

**Figure 7.** Model 3's PSIS diagnostic plot.

Diagnostic statistics and trace plots support the convergence of Model 3's Markov chains. The *n_eff* values are greater than 1000, and *Rhat* values equal 1 (see Table 4). The trace plots visually validate the convergence of Model 2's Markov chains by showing the fluctuation around central equilibriums (see Figure 8).

**Table 4.** Model 3's simulated posterior coefficients.

| Parameter | Priors Reflecting a Belief in the Association | | | | Priors Reflecting Disbelief in the Association | | | |
|---|---|---|---|---|---|---|---|---|
| | **Mean** | **SD** | **n_eff** | **Rhat** | **Mean** | **SD** | **n_eff** | **Rhat** |
| *Constant* | 3.05 | 0.03 | 11,369 | 1 | 3.08 | 0.03 | 10,821 | 1 |
| *FormalFinInfo* | 0.19 | 0.04 | 12,047 | 1 | 0.13 | 0.04 | 11,046 | 1 |
| *InformalFinInfo* | 0.19 | 0.04 | 11,912 | 1 | 0.13 | 0.04 | 11,721 | 1 |

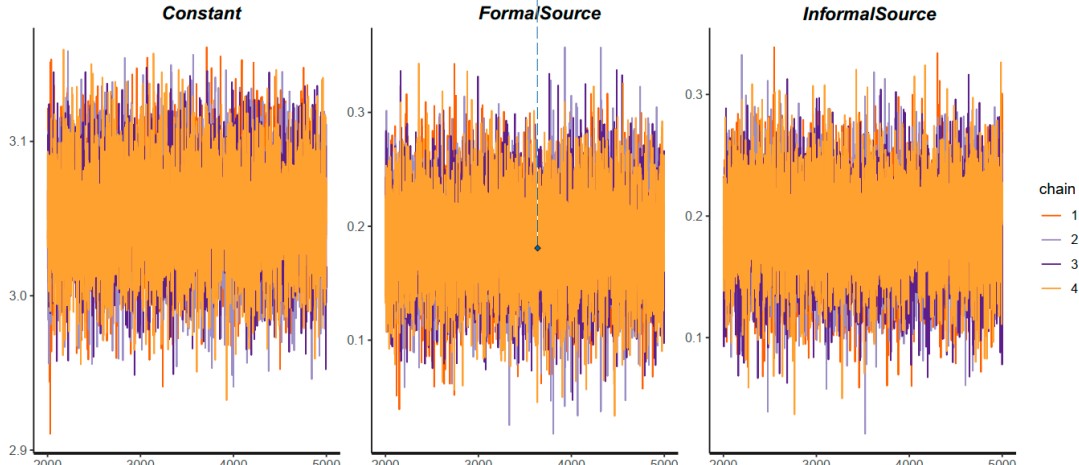

**Figure 8.** Model 3's trace plots.

The effects of coefficients simulated in Model 3 are consistent with those in Model 2. Specifically, both factors—accessibility to formal and informal financial sources—are positively associated with the general financial knowledge of the respondents ($\mu_{FormalFinInfo} = 0.19$ and $\sigma_{FormalFinInfo} = 0.04$; $\mu_{InformalFinInfo} = 0.19$ and $\sigma_{InformalFinInfo} = 0.04$). All coefficients' probability distributions are illustrated in Figure 9, with the highest posterior density intervals (HPDI) at 90%. Both *FormalSource*'s and *InformalSource*'s effects are highly reliable as their 90% HPDI are entirely located on the positive side of the *x*-axis's origin. Even when the priors are modified, the positive effects remain robust.

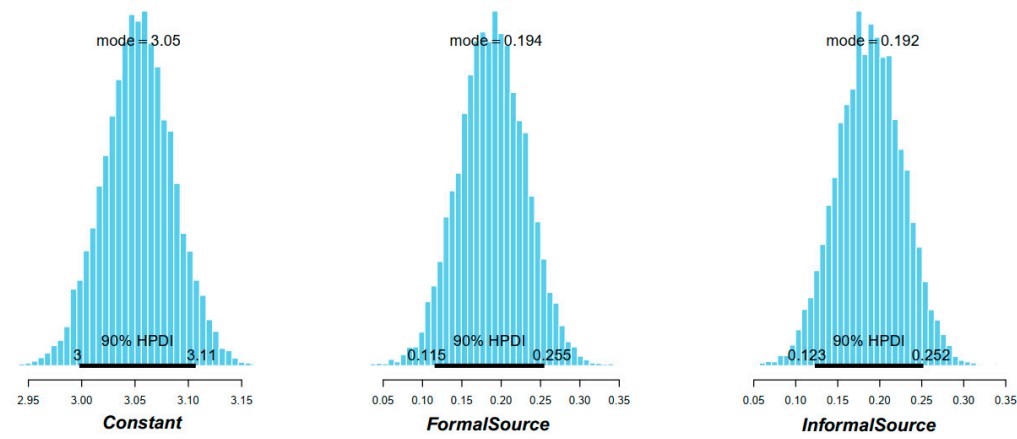

**Figure 9.** Model 3's posterior distributions on the density plot.

## 5. Discussion

The current paper is one of the first studies examining financial resilience through the lens of an information-processing perspective. It also provides the first empirical evidence of the relationships between financial information accessibility, financial literacy, and financial resilience.

Employing the BMF analytics on 839 Vietnamese individuals, we find that people with better financial knowledge and investment skills are less likely to be financially affected during the peak of the COVID-19 pandemic. Our research findings are consistent with studies of Clark et al. (2021) and Kurowski (2021), which also imply that more financially literate individuals are less at risk of facing financial difficulties due to the COVID-19 pandemic. This can be because less-literate people tend to struggle with saving and investing (Mandell and Klein 2009), eventually resulting in higher levels of bankruptcy, defaults, and foreclosures (Gerardi 2010). Notably, although individuals' investment skills are less financially affected by the COVID-19 pandemic, the finding is weakly reliable. This can be explained by the fact that the COVID-19 crisis affected the economy as a whole, leading to adverse effects on most types of investment. For that reason, being skillful in investment could not help much in buffering individuals from economic shocks induced by the pandemic. Another reason is that people who are financially literate tend to participate in stock market, and some have lost their financial liquidity due to economic downturns (Alessie et al. 2007), while people with lesser financial literacy were less likely to invest in the stock market and thus less likely to lose money during the financial crisis (Bucher-Koenen and Ziegelmeyer 2011).

Our findings highlight the importance of improving financial literacy among residents, as financial literacy is linked to financial resilience through more effective borrowing, saving, and spending patterns (Mitchell and Lusardi 2015). Russia, for example, supports responsible and rational financial behavior by providing financial education at the federal and provincial levels, which helps endogenize the nation's population to have a greater literacy level than the global average (Xu and Zia 2012). This made individuals less likely to experience a negative income shock during the global economic recession (Klapper et al. 2013). Financial literacy has also garnered increased attention in Vietnam due to the gradual shift of retirement and insurance planning duties from the public sector to the people (Jaax 2020; Schaumburg-müller 2005). Such a shift requires individuals to undertake more financial decisions, so the demand for financial outreach is also growing in the public sphere.

Our analysis also indicates that access to financial information through formal and informal sources can improve individuals' financial knowledge and investment skills. Specifically, people with a household member working in the financial sector and participating in financial courses/training tend to have more financial knowledge and better investment skills. Logically, people working in the financial sector must be financially literate in general or specific aspects. Enterprises in the financial sector usually invest in regular training of their employees for better financial knowledge and skills, such as capital management and financial contingency planning. As a result, household members working in the financial sector are likely to share such knowledge and skills with their families, increasing other household members' financial knowledge and investment skills. Meanwhile, financial training can also enhance one's financial knowledge and skills because financial education and training are designed to equip participants with financial knowledge and skills, such as saving, management, and investment.

The associations between financial resilience, financial literacy, and accessibility to formal and informal financial information sources validate two fundamental principles of the mindsponge theory: (1) outputs of the mental processes (e.g., responses to financial shocks) are influenced by the information existing within the mind (e.g., financial literacy), and (2) information existing within the perceivable range of the mind (e.g., accessibility to financial information) is likely to be absorbed and stored in mind (e.g., financial literacy). Besides these two principles, the mindsponge theory still has other principles that can

be applied to study financial literacy and financial resilience. For example, the subjective cost-benefit judgment towards information is a promising aspect for further study (Nguyen et al. 2021), as it helps delve into how a person can develop and use financial literacy effectively to mitigate the impacts of financial shocks.

These findings, supported by information-processing reasoning, hint at potential ways to improve individuals' financial literacy and resilience through financial information exposure. Through managing financial information exposure (e.g., financial training campaigns), policymakers can help remedy the financial literacy gap and economic inequality between gender, geographical regions, races, and ethnicity (Al-Bahrani et al. 2019; Fonseca et al. 2012). For instance, the inequality created through the educational attainment gap can be mitigated by financial training tailored to suit different learning styles and educational backgrounds, as financial literacy has a stronger effect than educational attainment on wealth (Behrman et al. 2012). Thanks to technological advances, financial training and education through digital platforms promise ways to improve financial literacy and resilience (Asian Development Bank 2019).

Furthermore, the positive association between accessibility to the informal source of financial information and financial knowledge and investment skill suggest the existing spillover effect of financial literacy among residents. In the age of digitalization, such an effect can be bolstered by the information dissemination capacity of the digital environment (for example, through social media, mobile phones, etc.). The effect can have double-edged impacts on the households' financial literacy and resilience levels. On the one hand, the spillover effect of financial information can support the stability of the macroeconomy, especially to shore up economic downturns due to economic shocks. On the other hand, financial information from informal sources, especially from the internet, is usually not evaluated or regulated, so there can be a chance that the acquired financial information is inaccurate (or fake). For example, investment frauds through the digital network were frequently reported during and after the COVID-19 pandemic (Ministry of Information and Communications 2021; VNA 2022). In such cases, the spillover effect of financial information might lead to misinformation, false beliefs, and poor economic decisions on a large-scale and cause disastrous consequences for the economy (Ecker et al. 2022; Bryanov and Vziatysheva 2021; Vuong et al. 2022d). The financial information monitoring framework should be designed and implemented to prevent the potential negative consequences of the spillover effect of financial information (Vuong et al. 2022e). Moreover, raising financial knowledge early in life, like mandating financial education and financial training, is still effective in mitigating the spillover of inaccurate information. It is because, in a collective, well-financially-literate, people can act as information filters that reduce the spreading of misinformation and financial information (Vuong et al. 2022e).

The financial landscape is dynamic, transforming much faster than before. Developing and implementing a financial roadmap is complex, and the public and private sectors must work together to achieve financial resilience. Individuals' financial fragility can be reduced if financial institutions and government adopt measures to instill confidence and endogenize public knowledge about financial management techniques, especially during economic downturns (Lusardi and Mitchell 2014). Besides this, collaborative actions among levels of government are required to increase the availability and accessibility of financial information to the public, such as through training campaigns and educational programs.

The current study is not without limitations, so we present them here for transparency (Vuong 2020). Firstly, financial literacy and resilience were proxied by the respondents' self-administration, so there might be a certain deviation from their real financial knowledge, skills, and resilience levels. Secondly, random sampling and on-site interviews could not be conducted due to the lockdowns and social distancing during the COVID-19 pandemic. Although we tried to ensure the data representativeness while collecting the survey through the digital network, we suggest readers interpret our study's findings cautiously.

**Author Contributions:** Conceptualization, M.-H.N. and Q.-H.V.; methodology, M.-H.N., V.-P.L.; software, V.-P.L.; validation, Q.-H.V.; formal analysis, M.-H.N., V.-P.L., T.-T.L.; investigation, T.-T.L., Q.-L.N. and P.-T.N.; resources, Q.V.K.; data curation, Q.V.K., V.-P.L. and T.-T.L.; writing—original draft preparation, M.-H.N., Q.V.K., T.-T.L., Q.-L.N., P.-T.N.; writing—review and editing, M.-H.N., Q.-L.N., P.-T.N. and R.J.; visualization, M.-H.N. and R.J.; supervision, Q.-H.V.; project administration, Q.-H.V.; funding acquisition, Q.-H.V. All authors have read and agreed to the published version of the manuscript.

**Funding:** This research received no external funding.

**Institutional Review Board Statement:** Ethical review and approval were waived for this study because ethical approval is not required by our institutes for social survey research.

**Informed Consent Statement:** Informed consent was obtained from all subjects involved in the study.

**Data Availability Statement:** The datasets generated and/or analyzed during the current study are not publicly available but are available from the corresponding author upon reasonable request.

**Conflicts of Interest:** The authors declare no conflict of interest.

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
