# Peer review of "Mindsponge-Based Reasoning of Households’ Financial Resilience during the COVID-19 Crisis"

_jrfm, doi:10.3390/jrfm15110542_

Round 1

Reviewer 1 Report

since the covid-19 affected people's life for about three years, the authors provide interesting report on  the current study aims to the effects of financial literacy and accessibility to financial information on the financial resilience of Vietnamese households through the lens of an information-processing perspective  with 839 samples for the investigation. it's a good work and good for publication. 

Author Response

Thank you very much for your comments! Please see the attached file for our detailed responses.

Reviewer 2 Report

See the file attached 

Author Response

(The authors gave the same response as above.)

Round 2

Reviewer 2 Report

The author has accepted all my comments and has definitely improved the paper.

Well done! I would like to see the paper published in the journal.